# Multi-Steps Fragmentation-Ion Trap Mass Spectrometry Coupled to Liquid Chromatography Diode Array System for Investigation of Olaparib Related Substances

**DOI:** 10.3390/molecules24050843

**Published:** 2019-02-27

**Authors:** Alaa Khedr, Soad S. Abd El-Hay, Ahmed K. Kammoun

**Affiliations:** 1Department of Pharmaceutical Chemistry, Faculty of Pharmacy, King Abdulaziz University, P.O. Box 80260, Jeddah 21589, Saudi Arabia; akammoun@hotmail.com; 2Department of Analytical Chemistry, Faculty of Pharmacy, Zagazig University, 44519 Zagazig, Egypt; soadselem@gmail.com

**Keywords:** olaparib, degradation products, fragmentation pattern, liquid chromatography with diode array and mass spectrometric detectors

## Abstract

A high-performance liquid chromatography-diode array-mass spectrometric (LC-DAD-MS) method was developed and validated to investigate the related substances of olaparib (OLA) in bulk form. OLA was exposed to acid–base hydrolysis, boiling, oxidation with hydrogen peroxide, and UV light followed by LC-DAD-MS analysis. OLA and OLA-related substances were simultaneously and quantitatively monitored by DAD at 278 nm and triple quadrupole mass spectrometry (QQQ-MS). The investigated compounds were auto-scanned by an ion trap MS which applied positive and negative modes separately. The fragmentation pathway was confirmed by applying multi-steps fragmentation to identify the resulted cleaved ions and their parent ion. OLA was found to be sensitive to the alkaline hydrolysis and less sensitive to UV light. Two major hydrolytic degradation products, including the protonated molar ions *m*/*z* 299 and *m*/*z* 367, were identified. Three potential impurities were also characterized. The LC-MS limit of detection (LOD) and limit of quantification (LOQ) were 0.01 and 0.05 ng/µL, respectively. The quantitative results obtained by LC-DAD was comparable with that of LC-QQQ-MS. The proposed method shows good intra-day and inter-day precision with relative standard deviation (RSD) <2%.

## 1. Introduction

Olaparib, or 4-[[3-[[4-(Cyclopropylcarbonyl)-1-piperazinyl] carbonyl]-4-fluorophenyl] methyl]-1(2*H*)-phthalazinone, was approved in December 2014 in the European Union [1] and the United States of America [2] as a targeted therapy for the treatment of breast cancer and mutated ovarian cancer. Olaparib (OLA) is a poly ADP-ribose polymerase (PARP) inhibitor. OLA inhibits the role of hereditary mutated breast cancer 1 (BRCA1) and breast cancer 2 (BRCA2) genes, which are involved in some ovarian, breast, and prostate cancers [3]. A survey of the literature showed that few methods have been reported for the analysis of OLA in biological fluids. A liquid chromatography-mass spectrometric method was found in the literature describing the quantification of OLA in either bulk form, in human plasma, or in combination with other drugs [4,5]. Thummar et al. (2018), described a validated stability indicating assay method of OLA using the high-performance liquid chromatography-diode array mass-spectrometric (LC-DAD-MS) method, however, the related substances, including potential impurities, were not considered [6]. The process for the preparation of OLA, according to the published patents, showed many intermediates with a percentage yield of the final product of not more than 96% [7,8].

The safety of a drug product is dependent on the level and toxicity of related substances that may arise from the active material [9]. Reporting the content threshold of drug impurities, as mentioned by International Conference on Harmonization (ICH) guidelines, is not more than 0.05% for unidentified materials, and not more than 0.1% or 1.0 mg per day intake, for a maximum dose of ≤2 g/day [9]. Recently, many articles have been published describing the importance and methods commonly applied for testing the organic-related impurities of active materials and products using LC-MS [10,11,12,13,14,15]. The labeled content of commercially available standard OLA is >98%. This leads us to the interest to test the purity of bulk OLA and its compliance with the International Conference on Harmonization (ICH) requirements regarding the limit of related substances [9]. So far, OLA is not yet official in any compendia, so it would be very important to develop a sensitive method to separate, characterize, and quantify all the degradants and process-related impurities that could be present in bulk drug material to assure the safety and quality of OLA. In this work, a liquid chromatography-diode array-mass spectrometric method was developed to investigate the purity of OLA and to characterize the possible related substances that could be released through exposure to different environmental variables. The method was optimized and validated according to the FDA and ICH guidelines [16,17].

## 2. Results and Discussion

The investigation of the related substances that might be released from OLA through exposure to environmental variables would be necessary as required by ICH guidelines [9]. The electrospray ionization mass spectrometry (ESI-MS) response factor of OLA matched with its related substances were identical and showed the same quantitative results as LC-DAD measurements. This might be attributed to the existence of the same ionization targets on all related substances. The non-DAD detectable impurities have showed enough sensitivity in ESI-MS for quantitative calculation. The potential impurities and degradation products of OLA could be simultaneously quantified by DAD measured at 278 nm and ESI-QQQ-MS. The positive and negative IT-MS analysis of forced degraded samples enabled the characterization of OLA-related substances, applying multi-steps auto-fragmentation mode (± auto MS^n^, *n* = 3).

### 2.1. Optimization of Chromatographic and QQQ-MS Conditions

Five degraded OLA sample solutions were prepared to account for the effects of acid and base hydrolysis, as well as the effects of heat, oxidation, and light. Several chromatographic conditions were applied and optimized to achieve the best resolution and detection. These samples were analyzed using LC-DAD-MS. These modifications included the effects of the column type, the mobile phase composition, and the settings of the QQQ-MS and IT-MS ion optics. The optimal chromatographic and MS conditions were achieved as described in the experimental section. The optimal mobile phase composition was acetonitrile: 6.5 mM ammonium acetate with 0.01% formic acid (*w*/*v*) in water (30:70, *v*/*v*), which enables the complete separation of OLA from the generated degradation products (DPs) and process-related impurities in a total run time of 25 min using the DAD detector. The ESI-MS detector was sensitive enough to trace low levels of DPs and process-related impurities. The Nucleodur 100–5 C18 end capped column was suitable for optimal separation of analytes, however, other HPLC columns packed with C18, five microns, showed unacceptable resolution of OLA from IMP-B and IMP-C. In addition, upon using methanol as an organic modifier in the mobile phase, a progressive tailing of all analytes was observed along the run. Figure 1 showed the generated HPLC-DAD chromatograms of OLA-forced degradation at different stress conditions.

### 2.2. Forced Degradation of OLA

Simultaneously, the DAD eluted compounds were monitored by IT-MS, applying positive and negative modes separately with the auto-MS^n^ function (Figure 2). A relatively concentrated standard solution (500 ng/μL) was analyzed to detect OLA-related substances. The LC-DAD-MS data of standard OLA and the heated solution (0.5 mg/mL) at 90 °C showed three minor foreign well separated peaks from the principle compound (Figure 1a and 2a). The three foreign compounds were characterized by +IT-MS and +QQQ-MS. These compounds were considered as potential impurities, labeled as IMP-A, IMP-B, and IMP-C and assigned to molar ions [M + H]^+^ of, *m*/*z* 409 at 13.6 min, *m*/*z* 417 at 19.5 min, and *m*/*z* 326 at 20.3 min, respectively. The percentages of the detected impurities A, B, and C, in the bulk OLA solution, were 0.23, 0.02, and 0.09% (*w*/*w*), respectively (Table 1). OLA showed acceptable stability upon exposure to oxidation, heat, and acid hydrolysis. However, the base treated solution and UV-exposed OLA samples showed many degradation products, and the remaining amount of OLA, as a percentage, ranged from 80 to 65% (Figure 1 and Figure 2, Table 1). The molar ions, [M + H]^+^ of the two DPs, released due to base catalyzed hydrolysis, were *m*/*z* 299 at 6.5 min (DEG-A) and *m*/*z* 367 at 11.0 min (DEG-B). The generated percentage concentration of both DEG-A and DEG-B were 20.8 and 13.51%, respectively and considered as potential degradation products (Table 1). Meanwhile, the relative amounts of all OLA-impurities B and C were not detected after base-catalyzed hydrolysis or UV-exposure as shown in Figure 1 and Figure 2, and Table 1. The concentration of the remaining OLA in all stress testing experiments was determined after dilution 10-fold (50 ng/μL) for LC-DAD analysis and 100-fold (5.0 ng/μL) for LC-QQQ-MS analysis.

### 2.3. Identification of Related Substances

The LC-DAD-MS system was programmed to monitor the product ions applying a fragmentation voltage of 140 V and collision energy voltage of 25 V for the pre-defined molar ions at the specified elution time. Moreover, all prepared OLA samples, including degraded samples, were analyzed by positive and negative ion trap MS applying auto-MS^n^. The fragmentation procedure enabled us to follow the most abundant MS^2^ fragment and gave more confident results in structure confirmation. The priority rule and fragmentation pattern were first defined for the principle compound (OLA) then applied to other MS spectra corresponding to OLA-related substances. Alternatively, the negative MS^2^ and MS^3^ showed the resultant negative fragments using the same priority rule set for +MS^n^. The average +MS^2^ spectra were extracted and subtracted from the average baseline response (0.20 to 1.20 min).

The +IT-MS^2^ spectrum of OLA, *m*/*z* 435, [M + H]^+^, was characterized by the most abundant peaks at *m*/*z* 367 (100%) (a), *m*/*z* 281 (20%) (b), and *m*/*z* 324 (5%) (c) (Figure 3). The product ion at *m*/*z* +367 (100%) was generated due to cleavage of “cyclopropane carbonyl” moiety from OLA, [M − 69 + 2H]^+^ and another abundant fragment ion at *m*/*z* +281, [M − 153]^+^, is assigned to the cleavage of “cyclopropyl(piperazin-1-yl)methanone” moiety. The fragment ion at *m*/*z* +324, is assigned to [M − cyclopropane carbonyl − (NHCO) + 2H]^+^. The most abundant MS^2^ fragment was automatically selected for further fragmentation to generate auto-MS^3^ spectrum. The +MS^3^ spectrum of *m*/*z* 435→367 showed protonated ions at *m*/*z* 281 (100%), 324 (28%) and 233 (7%). The negative MS^2^ spectrum of OLA, *m*/*z* 433 [M − H]^−^, showed an abundant ion at *m*/*z* 253 (100%) (a’), as shown in Figure 3, and *m*/*z* 233 (60%) (b’) due to further loss of the fluoride atom. Moreover, the -MS^3^ spectrum of *m*/*z* 433→253 ion showed an abundant fragment ion at *m*/*z* 210 (100%) due to the loss of NHCO moiety (Appendix A). The molar protonated ions and its related substances, including degradation products, were characterized by QQQ-MS and IT-MS separately. Approximately matched MS^2^ spectra generated by +IT-MS were obtained by +QQQ-MS applying a collision energy voltage of 25 V.

Similarly, the fragmentation pathway of the chemical structures of released DPs were identified, as shown in Figure 4. All characterized related substances showed the same fragmentation order and pattern as the principle compound, using ±IT-MS^2,3^ and -QQQ-MS^2^. Figure 4 showed the characterized product ions (+MS^2^) of selected molar ions monitored by +QQQ-MS. The use of IT-MS was more helpful in the characterization of related substances due to the trapping option that enables tracing of the source of generated fragments using ±MS^2^ and ±MS^3^ scan modes. The degradation pathway of OLA, IMP-A, IMP-C, and DEG-B is preferably proceeded via the formation of *m*/*z* 299, as shown in Figure 5. Samples exposed to stress conditions showed either no or a very low level of IMP-A due to the formation of DEG-A (Table 1).

### 2.4. Method Validation

#### 2.4.1. Linearity

A linear relationship between the peak area and the concentration of OLA over a range of 0.5–200 ng/μL and 0.05–10 ng/μL was found for DAD and +QQQ-ESI-MS measurements, respectively. The regression coefficient of both calibration curves was 0.9998. The response factor (slope) of both DAD and +MS detection methods were 6.705 ± 0.24 and 2,282,799 ± 4.94, respectively. The lower limit of quantitation (LOQ) was estimated by satisfying two criteria: The S/N ratio is not less than nine and the relative standard deviation (RSD) of five replicate injections of the LOQ solution is less than 6% [18]. The base line noise of total ion MS chromatogram was calculated as reported by Dong et al. [19,20] The limit of detection (LOD) of the DAD and +MS methods were 0.1 ng/μL and 0.01 ng/μL, respectively, whereas the limits of quantification (LOQ) were 0.5 ng/μL and 0.05 ng/μL, respectively.

#### 2.4.2. Precision and Accuracy

The precision of the developed method was estimated as a function of the RSD values of the determined amount of OLA (within the calibration range) using UV detection. Intra-day precision was assessed by injection of the standard solution at three concentrations three times during a day. The same was done for the inter-day precision test except that the injection of the samples was every day for three days. Results show that the RSD values for both the intra- and inter-day precisions did not exceed 2%.

#### 2.4.3. Accuracy

Known amounts of OLA were spiked in (added to) water to give claimed final concentrations of 0.5, 5.0, 20.0, and 100.0 ng/μL. The percentage error of the found concentrations calculated from the LC-DAD data did not exceed 0.71%. The percentage of error was calculated by subtracting the spiked amount from the found amount divided by the spiked amount and multiplied by 100. The RSD of the recovered amounts did not exceed 1.22%.

#### 2.4.4. System Suitability and Selectivity

The LC-DAD chromatographic performance parameters, including the RSD of the retention time of OLA (t_R_, min 23.5 ± 0.16 min), capacity factor (k’), selectivity coefficient (α), resolution (R), and the USP theoretical plates (N), were found to be within the acceptable ranges, as shown in Table 2. The selectivity was evaluated using the DAD and MS chromatograms. The peak corresponding to the standard OLA was DAD-scanned at the apex, upslope, and downslope and normalized by estimating the matching percentage with the corresponding peak separated from the DAD-degraded sample. The peak purity index of OLA monitored using the DAD detector was 100 ± 0.5%. Furthermore, the ESI-MS recorded [M + H]^+^ chromatograms of the same sample and the standard solution showed no peak overlapping. These data confirm the suitability and selectivity of the method (Figure 1 and Figure 2).

#### 2.4.5. Robustness

To evaluate the robustness of the method, one chromatographic parameter was changed while the other parameters remained constant. A standard solution (claimed, 0.5, 1 mg/mL). The robustness of the method was tested after changing the mobile phase composition. Ammonium acetate, 0.5 g/L, was added to obtain a more precise retention time. The amount of formic acid used, 100 μL/L water, was used to improve positive ESI ionization at both positive and negative MS polarities.

## 3. Materials and Methods

### 3.1. Chemicals

Olaparib (Purity ≥98%) was purchased from LKT Laboratories (St. Paul, MN, USA). All reagents and solvents used were of HPLC grade. Ultrapure water was generated using a Milli-Q water purification system. Solvents used for the preparation of standard solutions were bubbled with nitrogen gas (99.999%) to expel dissolved oxygen gas.

### 3.2. Preparation of Calibration Solutions

The stock solution of OLA was prepared by dissolving 10 mg of OLA in 1.0 mL of dimethyl sulfoxide then diluting to 10 mL with acetonitrile. Two serial dilutions were prepared in acetonitrile spanning a range of 1.0 to 100 ng/μL, and 0.05 to 10 ng/μL, to construct the LC-DAD and LC-MS calibration curves, respectively. A volume of 5 μL from the stock solution was injected for LC-DAD-MS analysis, to estimate the material purity with the aid of UV-scans and positive molar ions of eluted peaks, simultaneously.

### 3.3. Liquid Chromatography-DAD and Mass Spectrometry

An Agilent 6320 liquid chromatography-ion trap mass spectrometer (LC-IT-MS) and Agilent (Palo Alto, CA, USA) 6420 liquid chromatography-triple quad mass spectrometer (LC-QQQ-MS) were used for the characterization and quantification of OLA and its related substances, respectively. Each MS system was connected to an HPLC-Agilent 1200 system equipped with an autosampler, a quaternary pump, and a column compartment (Palo Alto, CA, USA). Both systems were equipped with ChemStation software (Rev. B.01.03 SR2(204)). The IT–MS was controlled using a 6300 series trap control version 6.2 Build No. 62.24 (Bruker Daltonik GmbH, Billerica, MA, USA), and the general MS adjustments were: capillary voltage, 4000 V; nebulizer, 37 psi; drying gas, 12 L/min; desolvation temperature, 350 °C; ion charge control (ICC) smart target, 200,000; and max accumulation time, 200 millisecond (ms). The MS scan range was 50–650 *m*/*z*. Auto-MS^n^, positive and negative polarities were applied using the same mobile system. This system was used to characterize OLA and its related substances. The QQQ-MS system was controlled by MassHunter software (version B.03.01, Build 3.1.346.0). The QQQ-MS conditions were as follows: gas temperature, 330 °C; gas flow rate, 11 L/min; capillary voltage, 4000 V—and nebulizer: pressure, 35 psi; fragmentor voltage, 135 V; cell accelerator voltage, 7 V; and +MS scan range, 50–650 *m*/*z*.

The extracted ion chromatographic peaks were quantified by QQQ-MS parallel with DAD-chromatograms retrieved at 278 nm. The separation was performed on a Nucleodur 100–5, C18ec column (4.6 × 250 mm, 5 µm, Macherey-Nagel, Düren, Germany) maintained at 25 °C. The analytes were eluted using a mobile system composed of acetonitrile: 6.5 mM ammonium acetate with 0.01% formic acid (*w*/*v*) in water (30:70, *v*/*v*), and pumped at a flow rate of 0.5 mL/min. The screw-capped (PTFE/silicon) total recovery autosampler vials (1 mL, 12 × 32 mm) were purchased from Waters (Milford, MA, USA).

### 3.4. Forced Degradation

A concentrated standard OLA solution was prepared in acetonitrile, 1 mg/mL, and used for stress testing. The concentrations of the release-related substances were calculated simultaneously from the corresponding calibration curves of DAD and MS relative to the principle peak. All stressed samples were also re-analyzed simultaneously after dilution 10- and 100-fold to calculate the remaining intact OLA monitored by DAD and MS.

#### 3.4.1. Effect of UV-Irradiation

The UV-lamp used for photo-stress testing was 125 cm long (General Electric, Cincinnati, OH, USA). This lamp is installed in a Laminar flow cabinet (SterilGard Hood, Class II) and functions as a germicidal UV light source (The Baker Co. Inc., Sanford, Maine, USA). A sample film of OLA powder with a thickness of approximately 0.1 mm was placed in a watch glass dish and subjected to UV irradiation from a distance of 5 cm for 24 h. A claimed concentration of 500 ng/μL was prepared in acetonitrile and a volume of 5 μL was injected for LC-DAD-MS analysis.

#### 3.4.2. Effect of Oxidation

In a 2 mL brown color reaction vial, 500 μL of 1 mg/mL OLA solution was mixed with 500 μL 30% hydrogen peroxide (*w*/*w*), capped, left at 60 °C in a hot air oven for 30 min, cooled, vortexed, and transferred to an autosampler vial. A volume of 5 μL was injected immediately for LC-DAD-MS analysis.

#### 3.4.3. Effects of Acid and Base Hydrolysis

In a 2 mL brown color reaction vial, 500 μL of 1 mg/mL OLA solution was mixed with 250 μL 1 M hydrochloric acid, capped, left at 60 °C in a hot air oven for 30 min, cooled, mixed with 250 μL 1 M NaOH, vortexed, and transferred to an autosampler vial. Base catalyzed hydrolysis was carried out applying the same procedure but using 1 M sodium hydroxide instead of 1M HCl. A volume of 5 μL was injected for LC-DAD-MS analysis. The samples were reheated for 30 min and re-analyzed to check the reaction progress.

#### 3.4.4. Effect of Heat

In a 2 mL brown color reaction vial, a volume of 500 μL of 1 mg/mL OLA solution was mixed with 500 μL water, capped, left at 95 °C in a hot air oven for 30 min, cooled, vortexed, and transferred to autosampler vial. A volume of 5 μL was injected immediately for LC-DAD-MS analysis.

## 4. Conclusion

A sensitive and selective LC-DAD-MS method was developed and validated for testing the purity and stability of OLA in bulk form. The use of ±IT-MS^n^ and +QQQ-MS^n^ was a useful shortcut for the clear structural characterization of OLA-related substances. The MS fragmentation pathway of the principle compound enables tracing and identification of the generated OLA-DPs. The bulk OLA was found to contain three potential impurities including, IMP-A (*m*/*z* 409), IMP-B (*m*/*z* 417), and IMP-C (*m*/*z* 326). OLA is very sensitive to the alkaline matrix and generates two major hydrolytic products including DEG-A and DEG-B. Further toxicological examinations are needed to test the impact of OLA potential impurities. Moreover, the product containing OLA should be stored away from light and protected from packaging material that might leach alkali metals. The related substances existed at low concentrations, beyond the LOD-DAD level, could be identified and quantified by LC-MS. Both, LC-DAD and LC-MS measurements could be used equally for quantitative analysis of DPs.

## Figures and Tables

**Figure 1 molecules-24-00843-f001:**
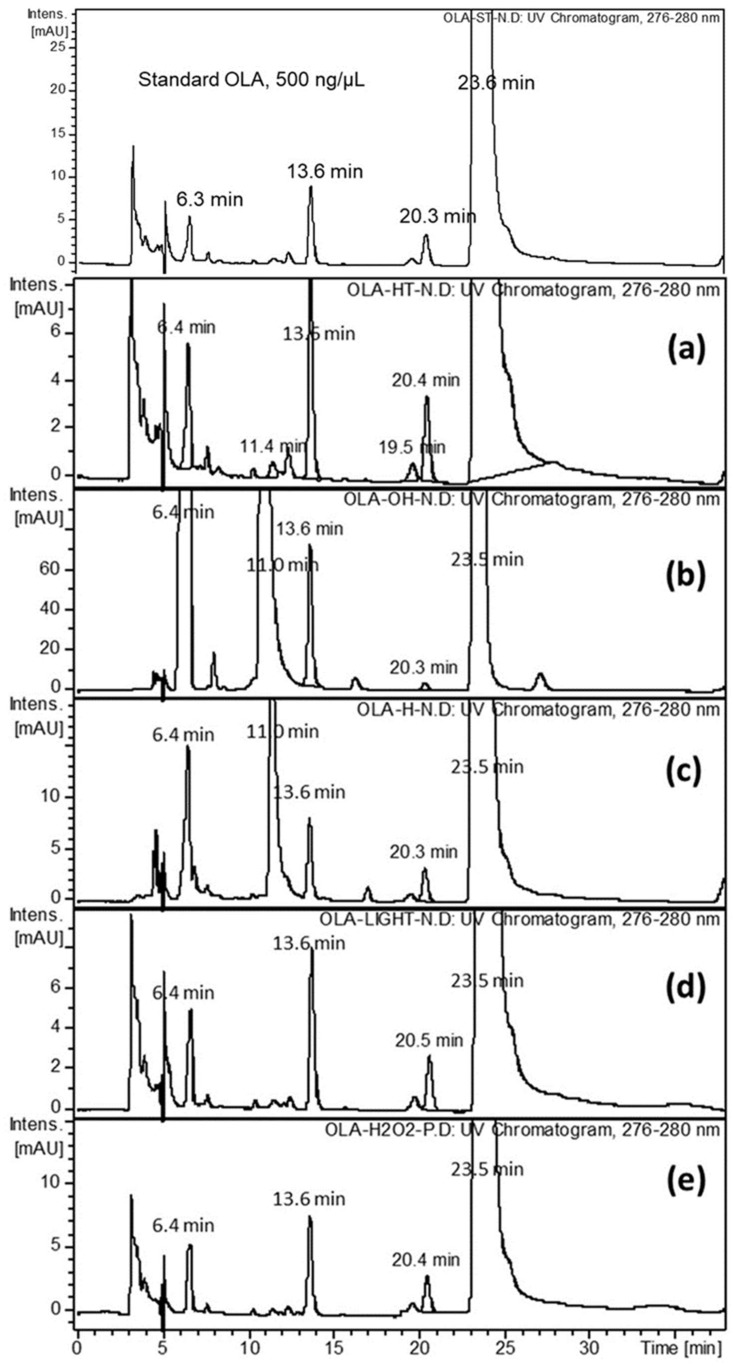
Representative LC-DAD chromatograms of standard olaparib at 23.5 min, 500 ng/μL, versus; heated in water at 90 °C (**a**), heated in 1 mol/L NaOH (**b**), heated in water 1mol/L HCl (**c**), exposed to UV light (**d**), and oxidized with H_2_O_2_ solution (**e**).

**Figure 2 molecules-24-00843-f002:**
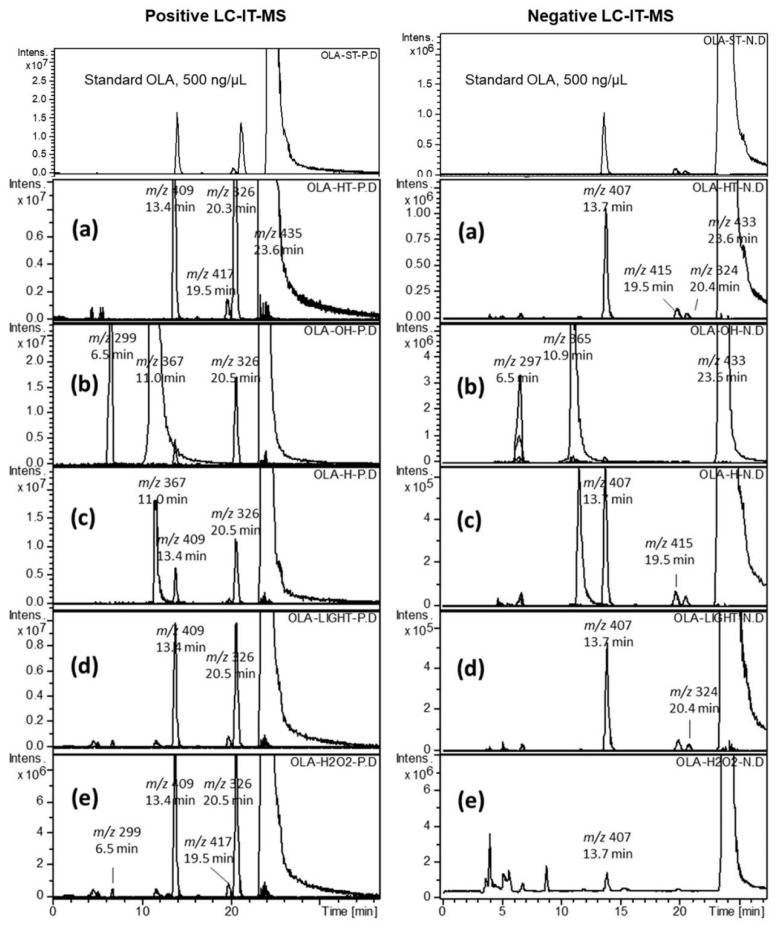
Extracted positive and negative MS ion chromatograms of standard olaparib, 500 ng/μL, versus; heated in water 90 °C (**a**), heated in 1 mol/L NaOH (**b**), heated in water 1 mol/L HCl (**c**), exposed to UV light (**d**), and oxidized with H_2_O_2_ solution (**e**).

**Figure 3 molecules-24-00843-f003:**
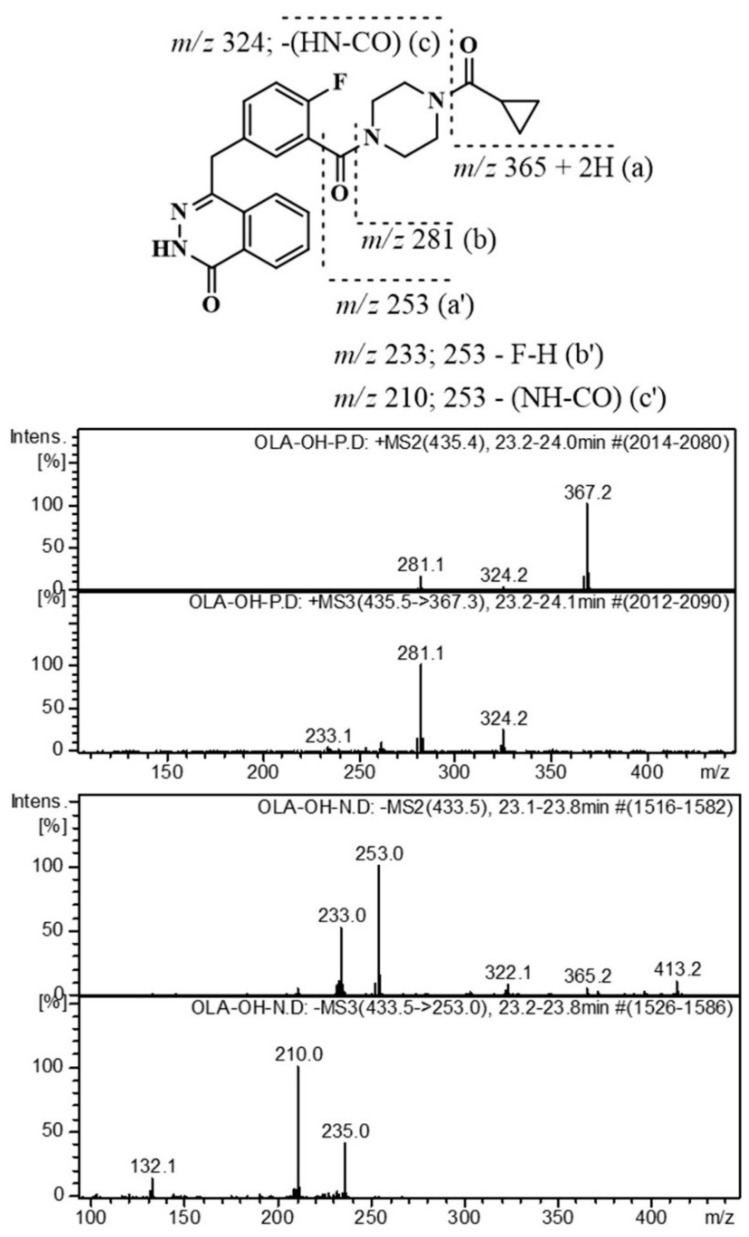
Positive and negative IT-MS^2^ and IT-MS^3^ spectra of olaparib.

**Figure 4 molecules-24-00843-f004:**
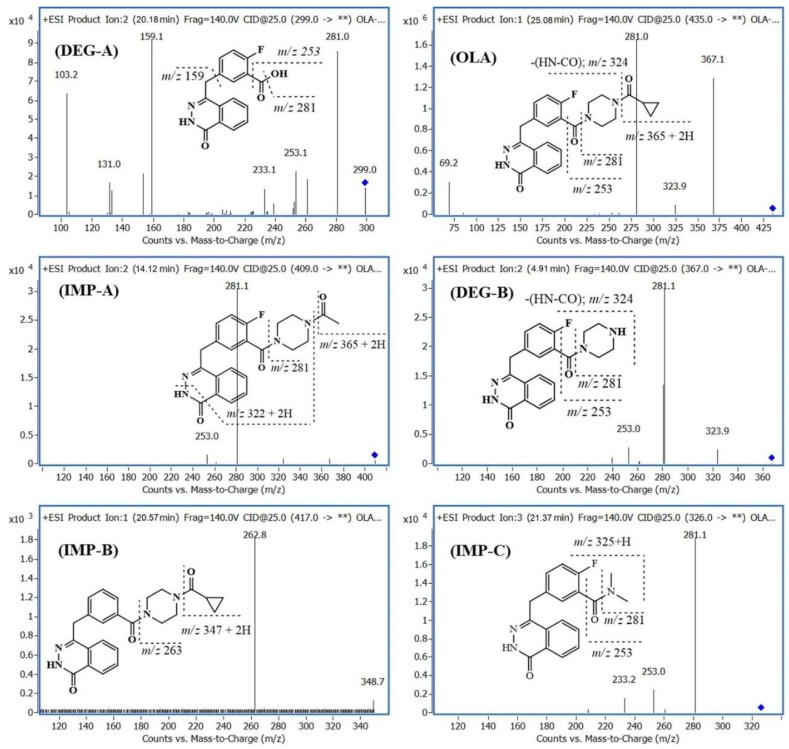
Average positive product ion spectra (QQQ-MS^2^) of olaparib, olaparib impurities and its degradation products.

**Figure 5 molecules-24-00843-f005:**
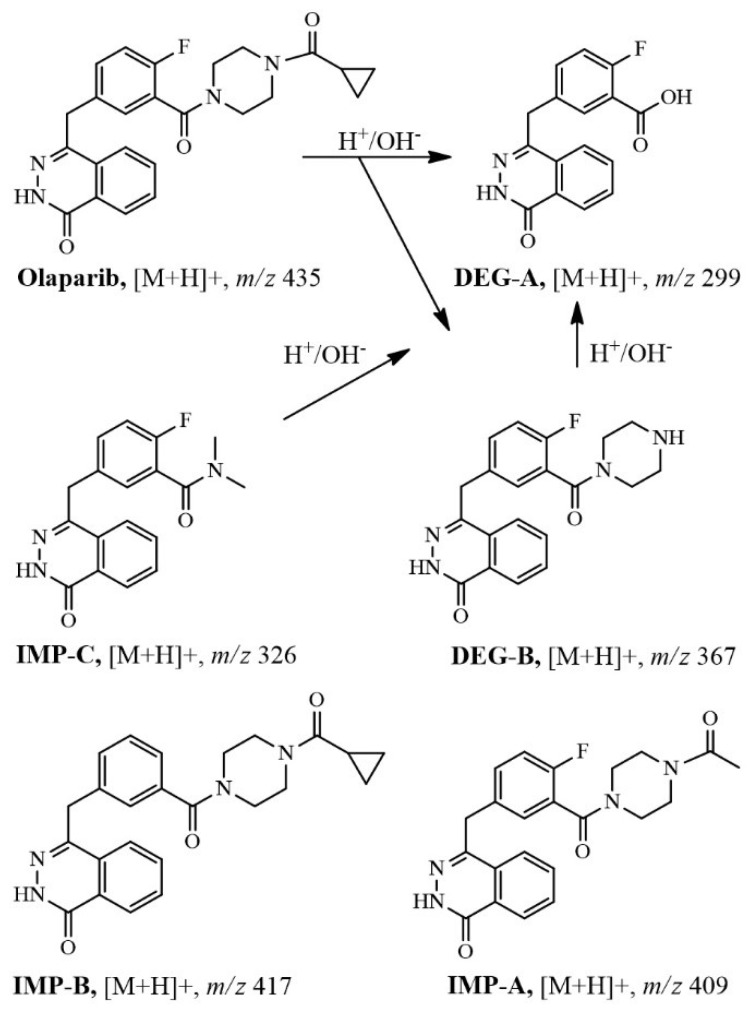
MS fragmentation pathway of olaparib.

**Table 1 molecules-24-00843-t001:** Calculated percentage amounts of olaparib and olaparib-related substances monitored by DAD (278 nm) and +QQQ-MS, simultaneously.

Name	[M + H]^+^ *m*/*z*	Rt, min	Bulk, % (*w*/*w*)		Heat, % (*w*/*w*)		H_2_O_2_, % (*w*/*w*)		Acid, % (*w*/*w*)		Base, % (*w*/*w*)		UV, % (*w*/*w*)	
			DAD	MS	DAD	MS	DAD	MS	DAD	MS	DAD	MS	DAD	MS
DEG-A	299	6.5	0.03	0.04	0.33	0.32	0.81	0.83	0.30	0.29	20.85	21.22	9.40	9.31
DEG-B	367	11.0	0.00	0.00	0.11	0.09	0.0	0.0	1.20	1.21	13.51	13.62	7.30	7.61
IMP-A	409	13.6	0.23	0.16	0.24	0.23	0.19	0.17	0.20	0.20	0.00	0.12	0.00	0.06
IMP-B	417	19.5	0.02	0.13	0.14	0.16	0.0	0.0	0.02	0.02	0.00	0.00	0.00	0.00
IMP-C	326	20.3	0.09	0.21	0.11	0.13	0.10	0.10	0.10	0.10	0.00	0.00	0.00	0.00
OLA	435	23.5	99.15	99.21	99.00	98.87	98.20	98.15	97.78	97.38	65.02	64.30	81.24	81.02
% of total related substances			0.37	0.54	0.93	0.93	1.10	1.10	1.82	1.82	34.36	34.96	16.70	16.98

* The standard deviation of the computed percentage values did not exceed 1.58.

**Table 2 molecules-24-00843-t002:** Chromatographic parameters of olaparib analyzed by HPLC-DAD, n = 6 *.

Label	Retention Time, (min)	W_(0.05)_, min	USP Tailing	*k*	α	USP Resolution	USP Plate Count
DEG-A	6.5	0.42	1.00	1.3	2.2	15.5	3832
DEG-B	11.0	0.58	0.98	2.9	1.3	14.1	5755
IMP-A	13.6	0.57	1.02	3.9	1.5	22.7	9109
IMP-B	19.5	0.55	1.03	6.0	1.0	19.7	20112
IMP-C	20.3	0.52	0.97	6.3	1.2	17.3	24384
OLA	23.5	1.02	1.03	7.39			8493

* *k*, retention factor; and α selectivity coefficient.

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
