# Peer review of "Multi-Steps Fragmentation-Ion Trap Mass Spectrometry Coupled to Liquid Chromatography Diode Array System for Investigation of Olaparib Related Substances"

_molecules, 2019, doi:10.3390/molecules24050843_

Round 1

Reviewer 1 Report

The manuscript reports a high-performance liquid chromatography-diode array-mass spectrometric method to investigate the related substances of olaparib in bulk form, which has a certain novelty in comparison with other reported methods for olaparib analysis. There are some minor suggestions for improvement for final acceptance.

1. Line 14-15, olaparib was abbreviated in line 14, but was still presented in its full name? Check the whole manuscript.

2. Please provide a sentence to give the significance or background for the analysis of olaparib.

3. Materials and methods section should be exchanged with Results and discussion section.

4. Results and discussion part, more comparison with other methods in determining olaparib should be provided.

5. Line 143, I recommend give the description of calculation for LOD, LOQ, and other parameters such as precision and accuracy… Some published papers about method validation can be referenced and cited. 1) A novel approach for simultaneous analysis of perchlorate (ClO4) and bromate (BrO3) in fruits and vegetables using modified QuEChERS combined with ultrahigh performance liquid chromatography-tandem mass spectrometry. Food Chemistry 2019, 270, 196-203. 2) Modified QuEChERS purification and Fe3O4 nanoparticle decoloration for robust analysis of 14 heterocyclic aromatic amines and acrylamide in coffee products using UHPLC-MS/MS. Food Chemistry, 2019, 285, 77-85. 3) Modified QuEChERS combined with ultra high performance liquid chromatography tandem mass spectrometry to determine seven biogenic amines in Chinese traditional condiment soy sauce. Food Chemistry, 2017, 229, 502-508.

6. Figure 1, please indicate which peak is for olaparib in the figure?

7. Line 159, about recovery experiment, why not use sample solution to replace water for spiking? How about matrix effect?

8. Standards of Figure 1 and Figure 2 should be further improved.

Author Response

Dear Dr. Reviewer

This is to thank you for your efforts making this work more clear and valuable.

Find attached response.

Reviewer 2 Report

Comments to the authors:

The article describes the elaboration and validation of a LC-DAD-MS analytical method for the quality control of olaparib, a recently approved anticancer drug. The method is aimed to detect and quantify olaparib and its degradation products, appeared after acid-base hydrolysis, boiling, oxidation with hydrogen peroxide and exposure to UV light.

The article contains the description of the validation steps of the LC-DAD-MS and explains the structure of two main degradation products.

Some issues have to be addressed:

1. Optimization steps of the LC-DAD-MS method are unclear (which other mobile and stationary phases were used, which was the impact of temperature etc). In Line 77 it is stated that “Several chromatographic conditions were applied and optimized to achieve the best resolution and detection”, but no such conditions are explained.

2. Figures:

Legend of Fig 1 should clearly state it is a LC-DAD chromatogram.

The chromatograms (LC-DAD and LC-MS) of non-degraded olaparib should as well be represented in Fig. 1 and 2 in order to allow comparison with the degraded samples.

Figure 4 represents theAverage positive product ion spectra (QQQ-MS2) of abiraterone and its degradation products”. However, the article makes no reference to abiraterone. The authors should explain why the spectrum of this compound is inserted, and which is the relationship between abiraterone and IMP-A, IMP-B and IMP-C. Is the chemical nature of IMP-A, IMP-B and IMP-C a certain fact, or a potential possibility?

3. Several formulations in English are unclear.

Line 15:  Please correct ; to : Olaparib was forcedly degraded by subjecting to: acid-base hydrolysis....

Line 22:  Three potential impurities were also characterized – please define “characterized”. In the conclusion section they are referred to as “potential impurities” “IMP-A, IMP-B and IMP-C” without specifying their structure.

Line 37: “is also called” should be replaced with “is a”

Line 64: “The investigation of the related substances that might [be] released from olaparib would be necessary  to identify the molecular entities, the most effective environmental impact, and the quantities” - Please rephrase, the meaning of the sentence is unclear. 

Author Response

(The authors gave the same response as above.)

Round 2

Reviewer 1 Report

I have no other comments. The work can be accepted after improvement about method validation section with some references suggested.

Reviewer 2 Report

The authors performed the modifications requested by the reviewer.